# Low Loss Vertical TiO_2_/Polymer Hybrid Nano-Waveguides

**DOI:** 10.3390/nano13030469

**Published:** 2023-01-24

**Authors:** Isaac Doughan, Kehinde Oyemakinwa, Olli Ovaskainen, Matthieu Roussey

**Affiliations:** Department of Physics and Mathematics, Center for Photonics Sciences, University of Eastern Finland, P.O. Box 111, 80101 Joensuu, Finland

**Keywords:** optical waveguides, integrated optics, micro fabrication, polymer components

## Abstract

This article proposes a novel demonstration of a low-loss polymer channel hybridized with a titania core leading to a nano-waveguide elongated in the normal direction to the substrate. It is aimed at using the quasi-transverse magnetic (TM) mode as the predominant mode in compact photonic circuitry. A detailed design analysis shows how a thin layer of a higher-refractive index material in a trench within the core of the waveguide can increase the confinement and reduce the propagation losses. This thin layer, produced by atomic layer deposition, covers the entire polymer structure in a conformal manner, ensuring both a reduction of the surface roughness and a stronger field confinement. The trench can be made at any place within the polymer channel and therefore its position can be tuned to obtain asymmetric modal distribution. The waveguide is demonstrated at telecom wavelengths, although the material’s properties enable operation over a large part of the electromagnetic spectrum. We measured propagation losses as low as 1.75 ± 0.32 dB/cm in a 200 nm × 900 nm section of the waveguide core. All processes being mass-production compatible, this study opens a path towards easier integrated-component manufacture.

## 1. Introduction

A channel waveguide on a chip is composed of a core, a substrate, and a cladding. Except for rare exceptions, e.g., slot waveguides [1,2] and anti-resonant reflective waveguides [3,4], the core is usually constituted of one or several materials of a higher refractive index than the substrate and superstrate, ensuring the required light confinement. The refractive index difference between the three parts of the waveguide is a measure of the confinement strength of the propagating modes inside the core. Low refractive index contrast waveguides, such as glass waveguides made by ion exchange, usually have a large and high cross section to ensure good confinement of the optical modes within the core of the waveguide [5,6]. This is a synonym of low loss when the material constituting the waveguide is homogeneous and exempt from scatterers possibly perturbating the field propagation. However, in such a case, tight bends are prohibited to avoid extensive losses. On the other hand, nano-waveguides enabled by high refractive index contrast materials, such as silicon or germanium, allow very tiny and compact photonic circuitry [7,8,9]. Propagation loss is higher, but the reduced footprint of the components compensates for it. Polymers offer a trade-off between these two extremes since they have a refractive index ranging from 1.2 to 2 [10,11,12]. Moreover, direct electron beam lithography (EBL) is possible for some of them [13,14], and nanoimprinting is an interesting option for the fabrication of channel waveguides at large scale [15,16]. However, because of a refractive index close to that of the substrate, polymers may not be ideal for small-footprint devices.

The cross-sectional geometry of a waveguide is the other key parameter to tuning the effective index of a guided mode. In particular, the shape, size, and symmetry of the core will directly affect the birefringence, dispersion, confinement, and localization of a guided mode. Large core-based waveguides, usually the low refractive index contrast waveguides or the large 3 µm thick silicon waveguides, present modes with low dependence on the fine core modifications. In contrast, nano-waveguides are guiding modes reacting strongly to any tiny modifications such as a sidewall corrugation or a tapering.

We present, here, an alternative geometry allowing the use of polymer waveguides even at a small-footprint scale. The cross section of the waveguide is presented in Figure 1. The geometry, i.e., the cross section of the waveguide, consists of a polymer waveguide of width 2w+2t made of AZ-2070 (nLOF) electron beam resist of refractive index $n_{P}=1.601$ at λ=1550 nm [17,18,19]. A trench of width 2t is opened in the channel at a position $y_{T}$ varying from −a to a, with a=w+t and $y_{T}=0$ corresponding to the center of the waveguide. To ensure confinement of the field in the central core of the waveguide, the trench is filled up with a high-refractive index material. We opted for titania, TiO_2_ with $n_{{\mathrm{TiO}}_{2}}=2.24$ at λ=1550 nm, deposited by atomic layer deposition (ALD) in a conformal manner all over the polymer channel [20,21,22]. This fabrication method leads to a coating of thickness t on the sides and lateral unguiding regions of the waveguide, but also to a filling of the high aspect ratio trench, as sketched in Figure 1. The entire structure is fabricated on a thermally oxidized silicon wafer. The thermal oxide layer of 3 µm thickness ensures isolation of the optical mode with the silicon wafer. Therefore, a vertical high-refractive index nano-waveguide can be created.

In this article, we focus on the demonstration of the waveguide with a titania core. However, it is worth mentioning that as long as the refractive indices match the condition $n_{\mathrm{POLYMER}}<n_{\mathrm{TRENCH}}$, any changes in the materials are prevented. One can imagine a multitude of applications for which a hybrid waveguide is necessary but it is not easy at wafer scale. For instance, one can include a gain medium within the trench in order to create an on-chip amplifier [23]. Another application could be the use of quantum dots or dyes inside the trench to provide the up or down conversion of a signal in order to reach other wavelength ranges [24]. Emitters at very low concentration within the trench can provide very small amounts of photons and with a proper design of the polymer section, a single photon source could be achieved [25] for quantum photonics purposes.

## 2. Materials and Methods

### 2.1. Designing Tools

The finite-difference time-domain method (FDTD) is a tool for modeling and simulating nanoscale optical devices. It is well capable of handling dielectric structures that are inhomogeneous and complicated while producing field solutions that are well defined [26,27]. The waveguides studied in this work have been investigated with the commercial software, Ansys Lumerical suite. The configuration was set up as a structure group controlled with script as shown in Figure 1. The boundary conditions of the simulation region were set as perfectly matched layers (PML), which act as absorbers for any scattered field. The mesh step size used in the simulations was 20 nm in x, y, and z directions. This mesh size allows for numerical convergence of the results. The central illumination wavelength was 1550 nm. Due to high computational power and memory requirements with the 3D-FDTD, leading to time-consuming simulations, the waveguide and bend variation simulations were mostly carried out in 2D-FDTD using effective index approximation. The results were analyzed with MATLAB R2018b, MathWorks (Natick, MA, USA). The mode calculations were performed using the Lumerical in-built mode solver based on the finite difference method. In all calculations, the dispersion of the materials, measured by ellipsometry, was taken into account.

### 2.2. Fabrication Techniques

A layer of diluted negative resist (MicroChemicals AZ nLof 2070, Ulm, Germany) was spin-coated (Laurell WS-650MZ-8NPPB, Lansdale, PA, USA) at 1100 rpm on an oxidized silicon wafer. The polymer layer was soft baked at 110 °C on a hot plate for 1 min. The predefined waveguide structure was then transferred onto the resist layer by electron beam lithography (EBL, Vistec EBPG5000+HR, Dortmund, Germany). The acceleration voltage was set at 100 kV, and an exposure dose of 40 µC/cm^2^ was used. After exposure, the sample was baked hard at 110 °C for 1 min. The pattern was then developed in AR 300-47, AllResist (Strausberg, Germany), for 90 s and then stopped in de-ionized (DI) water for 30 s before rinsing with DI-water. nLof is a negative EBL resist, which means that exposed areas remain after development. The sample was further coated by atomic layer deposition (ALD) for a 100 nm titanium dioxide film. In the ALD process, the precursors were titanium tetrachloride (TiCl_4_) and water (H_2_O). The deposition rate was 0.0718 nm/cycle. The process temperature was set to 120 °C to ensure the deposition of an amorphous material.

Figure 2 describes the 4-step process flow for the manufacturing of the waveguides. The relatively large features of the structure would allow the use of nanoimprint technology instead of EBL for a faster, lower cost, and larger scale fabrication.

Several patterns were fabricated. This included bends of different radii of curvature (r) as well as long waveguides. These patterns were categorised into three chips as shown in Figure 3. Each waveguide had an input and output taper part of 3 µm width and 500 µm length. In chip 1, the bend radius was kept at 100 µm and the length L was varied from 500 µm to 5 mm. Chip 2 presents waveguides with varying bend radii from 2 µm to 20 µm whilst the length L was kept constant at 1 mm. In chip 3, the position of the trench filled by TiO_2_ was varied linearly within the bend for three different bend radii.

### 2.3. Characterization Methods

The final structure was characterized by the end-fire coupling method [28] described in the set-up shown in Figure 4. Light from a laser source was coupled through a tapered lens fiber into the waveguide and the transmittance power was measured using a power meter. The laser source used was a Tunics plus tunable laser source with wavelength accuracy of 1 pm. Its operation wavelength range was 1510 nm to 1640 nm. A polarization controller was used to set the desired polarization of light. The microscope objective (Olympus Plan N, 20×, NA = 0.4) was used to image the output of the waveguide on a near-infrared camera (Point Grey Chameleon CMLN-13S2M). The focused spot size of the lensed fiber used was 2 ± 0.5 µm. A Glan-Thompson GTH10M polarizer ensured the desired polarized light passed through to the NIR detector and power meter (Thorlabs S122c, PM100D) to measure the transmitted power. Alternatively, to the detector, the output mode could be imaged allowing a check at any time of the experiment that a well-coupling to the waveguide is performed without any modification of the measurement system. We used the cut-back method [29,30] to estimate the propagation losses by comparing transmissions through waveguides of varying lengths and fitting the length dependence under similar coupling conditions and surface roughness.

Different height lengths of waveguides were created for the propagation loss measurements. The transmission through each waveguide was measured five times, changing the injection side and the order of the measurement, i.e., from the longest to the shortest waveguide and vice versa. Results were averaged to obtain a single value used for the estimation of the propagation losses.

## 3. Results and Discussion

### 3.1. Optimization of the Waveguide Parameters

One figure of merit for many waveguide structures, especially as it relates to reducing propagation losses, is the mode confinement (Γ) within the core of the waveguide. To obtain the highest confinement, the parameters of the waveguide were optimized.

#### 3.1.1. Modal Effective Index

The optimization of the waveguide parameters was performed at the operation wavelength λ=1550 nm, i.e., the central telecom wavelength of the C-band, 1525 nm <λ< 1575 nm. The evolution with w, h, and t of the characteristics of the waveguides, such as the effective index, the confinement factor, and the birefringence between TE_00_- and TM_00_-modes were studied. The width and height of each polymer section were set to w=800 nm and h= 900 nm to ensure single modal behavior of the waveguide, when considering titania inside the trench. It should be noted that several solutions exist for the dimensions of the polymer sections to maintain a single mode waveguide. We opted for these dimensions for fabrication constraints and for a better mode confinement in the trench, for which the height plays the most important role. 

In Figure 5a,b, we show the dependence on the effective index of the two fundamental modes as a function of the TiO_2_ thickness 2t inside the trench. When 2t increased, both effective indices increased. For 2t> 200 nm, the waveguide became multimodal. Figure 5c,d shows the field intensity distribution for both fundamental modes. One can remark that the quasi-TM fundamental mode had a higher effective index and a better confinement in the TiO_2_-filled trench than the quasi-TE mode. While the quasi-TE mode was mainly confined in the polymer region on the interfaces TiO_2_/polymer ($\Gamma_{\mathrm{trench}}^{\mathrm{TE}}≃14\%)$, similar to standard nanowires, the fundamental quasi-TM mode was highly located inside the high refractive index region ($\Gamma_{\mathrm{trench}}^{\mathrm{TM}}≃39\%)$, similar to a 90°-rotated (vertical) standard ridge nano-waveguide. Moreover, the TM nature of the mode allowed a significant confinement factor ($\Gamma_{\mathrm{above}}^{\mathrm{TM}}≃5\%$) above the waveguide allowing for sensing with optimized parameters, e.g., a reduction of the height of the waveguide.

#### 3.1.2. Dispersion and Birefringence of the Waveguide

The dispersion, Figure 5e, in the waveguide was comparable to that observed in other waveguides, such as those used in silicon photonics, in the C-band [31]. Most importantly, the dispersion variation for the fundamental quasi-TM mode was relatively low over the studied wavelength range, which opens the path for broadband operating devices. The birefringence observed in Figure 5f was high (close to 0.1 refractive index unit). Such a difference between the modal effective indices of the modes would enable a polarization management in the waveguide for future applications, such as polarization rotation [32], or polarization maintaining waveguide [33], by acting on the shape, size, and position of the trench.

#### 3.1.3. Effect of the Position of the Trench

The TiO_2_ trench surrounded by polymer enabled a reduced bend loss and, therefore, a smaller bend radius. This comes from the fact that the trench yields a higher localization of the mode in the center of the waveguide, especially for the fundamental quasi-TM mode. The trench can be easily placed at any position within the waveguide leading to a tuning of the mode distribution central position. Figure 6 shows the TM_00_-mode profile ($\left|E_{x}\right|$) and the TE_00_-mode profile ($\left|E_{y}\right|$) along the y-direction as a function of the position $y_{T}$ of the trench. One can see that the mode distribution can be off centered with little or no modification of the overall modal distribution that remains well confined within the waveguide.

An important aspect to consider while designing a waveguide is the bend loss since bends are nearly unavoidable in most photonics circuitry. The tighter the bend, the smaller the footprint, and the higher the integration of the devices.

We performed simulations for three distinct positions of the trench, namely: $y_{T}=$ 0 nm, −400 nm, and −600 nm, keeping *w* = 800 nm, *h* = 900 nm, and *t* = 100 nm constant, and observed the transmission after a U-turn (180°) of the waveguide for different bend radii 1 µm < *r* < 20 µm. The results are shown in Figure 7. One can clearly see the expected trend for the transmission with an asymptotic limit for large values of r. In Figure 7a, the trench was shifted even on straight portions of the waveguide, while in Figure 7b, the trench was shifted only inside the bend portion. In the latter case, the titania part became elliptical to remain inside the polymer. One can remark that losses increased as soon as the trench was no longer centered; however, by operating a shift only in the bend part, losses were nearly negligible compared to the centered case. For the minimal radius r=1 µm, the loss for the bend was about 8% and rapidly decreased when r increased until reaching an asymptotic value below 1% for r>7 µm. Such a feature is a key towards waveguide coupling regions and other classical waveguide management that could be used in further applications. No major differences could be observed between the different core positions within the bend. This allows us to expect relaxed fabrication constraints and alignment, especially when considering nano-imprinting as a manufacturing method.

### 3.2. Fabricated Sample

The target parameters chosen for the fabrication of the structure were: *w* = 800 nm, *h* = 900 nm, and *t* = 100 nm. A less than 5% error in these dimensions was measured from the SEM pictures. Figure 8 presents several SEM pictures acquired on different samples.

Figure 8a is a scanning electron microscopy (SEM) picture of the cross section of the waveguide prior to atomic layer deposition (ALD) coating. The dimensions of the polymer-based channel were well respected despite the trench having a footing. Such an issue arises from limitations in the resist performance and fabrication. Figure 8b is an SEM picture of the cross section of the waveguide after TiO_2_ deposition (cleaved within the tapered section of the waveguide, which explains a larger waveguide than in Figure 8a). As expected, the material conformally coated the structure and a slight drop in the thickness was observable above the trench due to the unavoidable rounding of the edge of the waveguide [34] which was followed during the ALD process. The layer was thickened afterwards to fill this gap by an additional ALD coating of 10 nm. The roughness of the waveguide seems acceptable although one can see some contamination at the top of the waveguide both before and after the TiO_2_ deposition. Figure 8c is an SEM picture of the bent waveguides used for the loss measurements. The zoomed-in section on the bend highlights the precision with which the trench can be positioned within the waveguide. Among the tests we carried out prior to the fabrication of these waveguides, we realized that the exposure of the bottom part of the trench in the polymer channel is not obvious for all aspect ratios of the trench. We had to make a trade-off between the designed device and the fabrication possibilities, the most important of which was to ensure a single modal behavior of the waveguide; we chose to leave a partly unexposed resist inside the trench in order to keep the titania width as small as possible. Additional simulations have been performed including this default and no major change in the trends or even in the effective indices of the fundamental quasi-TM mode was observed. Finally, one should also note that such a bias is to be expected if the waveguide patterning is made by the nano-imprinting technique.

### 3.3. Experimental Determination of the Propagation Losses

The characterization setup used for measuring the propagation loss is shown schematically in Figure 4. In this article, we are mainly interested in demonstrating a waveguide for which the quasi-TM fundamental mode is predominant over the others. Therefore, the polarization was set to TM for all characterizations. Figure 9a shows an image of the quasi-TM fundamental mode at the output of one of the waveguides at *λ* = 1550 nm. We recall that tapers are used for the in- and out-coupling, which explains the size of the observed mode. To measure the propagation losses, several waveguides of different lengths with bend radius of 100 µm were measured while assuming that insertion losses are similar for all waveguides. Due to some damage (because of some fabrication, cleaving, handling issues), some waveguides were not measurable. Repeating the transmission measurements five times through the waveguides allowed us to give error bars on our measurements and give a better estimate of the propagation losses using the cut-back method from the slope of the curve presented in Figure 9b. The estimated propagation loss was measured to be 1.75 ± 0.32 dB/cm, which is close to the state-of-the-art for a titania nano-waveguide [35,36,37]. We would like to remark that due to the very thin additional re-coating of 10 nm of titania performed to fill the remaining gap, the waveguide became slightly multimodal (as Figure 5a predicted). However, this did not influence the loss measurements because we observed that the higher order TM mode was filtered out after the first bend.

In addition to the propagation losses, we measured the transmission through waveguides with varying bend radius. We observed constant transmission (with regard to an error similar to the one observed during the propagation loss measurements) down to 7 µm radius (as theoretically expected), followed by a sudden drop leading to non-measurable values.

## 4. Conclusions

In this paper, we have presented a detailed analysis of the geometric characteristics of a low loss TiO_2_/polymer waveguide. This structure is highly flexible in terms of operation over a large wavelength range. Furthermore, because such a waveguide is able to operate with very tight bends (radius < 7 µm), it offers a tradeoff between the quality of transmitted light, the device footprint, and relaxed fabrication and therefore can suit different applications. The trench feature enabled us to shift the position of the mode away from the center of the waveguide while still maintaining a high confinement without any modification to the geometry of the waveguide. Due to the confinement of the mode within the high index region, small footprint devices are achievable enabling mass production of compact photonic circuitry. The ALD fabrication process ensures a conformal deposition of a low-loss material inside the trench. These features allowed us to achieve waveguides with a low value for propagation loss (1.75 ± 0.32 dB/cm). The geometry of the waveguide itself allows high versatility in terms of the hybridization of materials. We have shown in this paper that despite a thin layer of high refractive index all over the sample, the mode remained confined inside the core of the waveguide. It yields an opening for many possible material combinations to be deposited with or without the conformal method. Finally, we point out that the use of polymer makes the core of the waveguide more robust and resistant to any contamination, mechanical stress and, therefore, such a concept could be used, for instance, in flexible photonics for wearable sensing devices.

## Figures and Tables

**Figure 1 nanomaterials-13-00469-f001:**
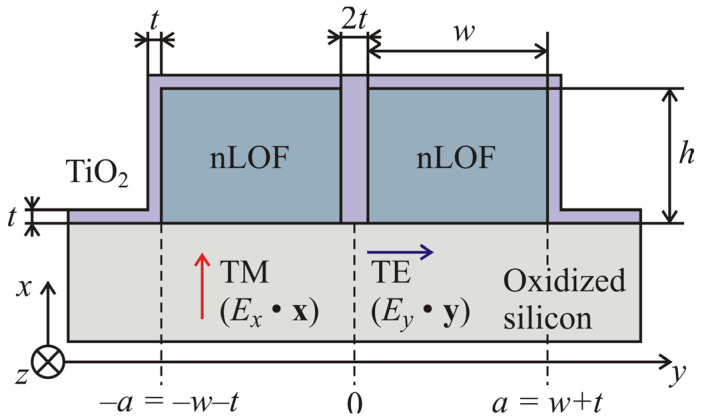
Cross-section geometry of the vertical nano-waveguide. The substrate is an oxidized silicon wafer. The polymer is AZ-nLof. The coating is TiO_2_. The coating thickness is t, the height and width of each part of the polymer region are h and w, respectively. The trench thickness is 2t and its position with respect to the waveguide is at $y_{T}=0$ when centered.

**Figure 2 nanomaterials-13-00469-f002:**
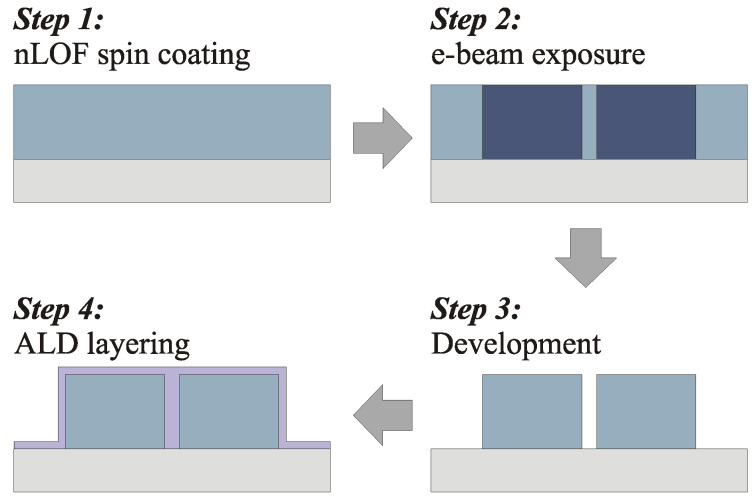
Process flow of the waveguide fabrication: resist deposition on an oxidized silicon wafer, electron-beam patterning of the nLof resist, development of the pattern, ALD coating of titania.

**Figure 3 nanomaterials-13-00469-f003:**
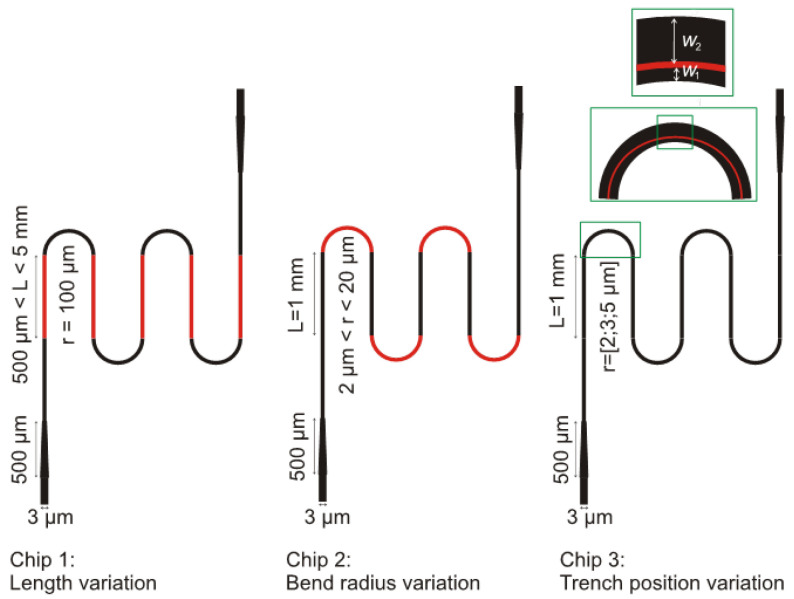
Layout of the chip. Chip 1: length variation from 500 µm to 5 mm. Chip 2: bend radius variation from 2 µm to 20 µm. Chip 3: TiO_2_ filled trench position variation W= 200 nm, 300 nm, 400 nm, and 500 nm for three bend radii of 2 µm, 3 µm and, 5 µm.

**Figure 4 nanomaterials-13-00469-f004:**
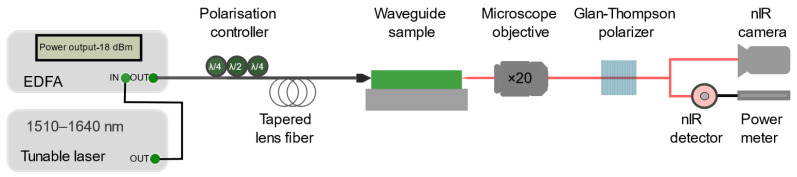
Schematics of the fiber characterization setup used to estimate the propagation losses in the waveguides using cut-back method. The detector of the power meter can be switched to a NIR camera for mode imaging.

**Figure 5 nanomaterials-13-00469-f005:**
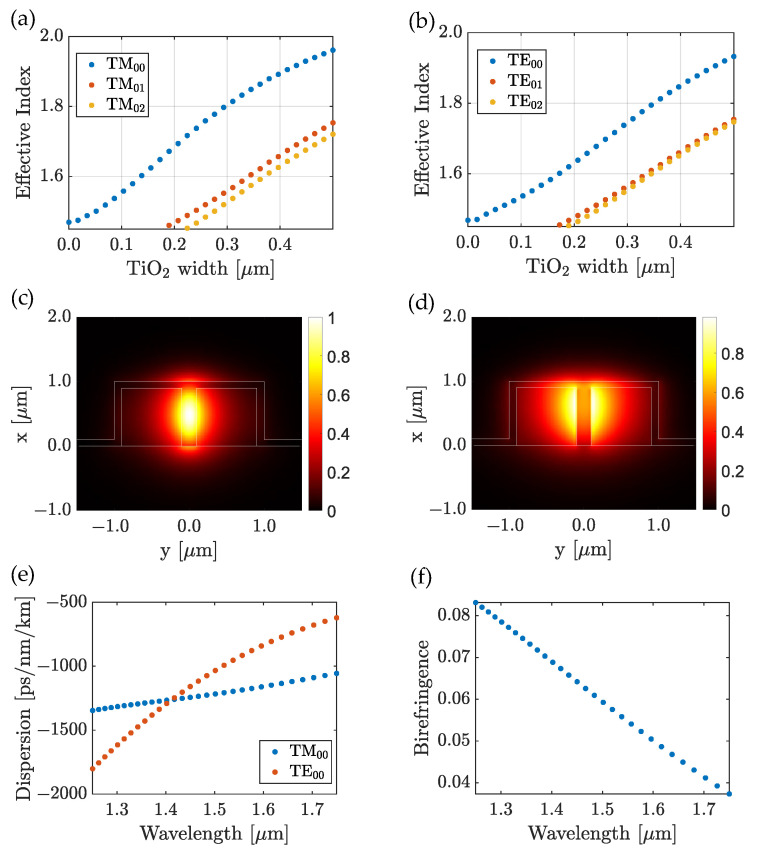
Waveguide structure design. (**a**,**b**) Effective index as a function of the TiO_2_ thickness 2t for quasi-TM and quasi-TE polarizations, respectively. (**c**,**d**) TM_00_ and TE_00_ mode distributions, respectively. (**e**) Dispersion of the TM_00_ (dotted red curve) and TE_00_ (dotted blue curve) modes. (**f**) Birefringence.

**Figure 6 nanomaterials-13-00469-f006:**
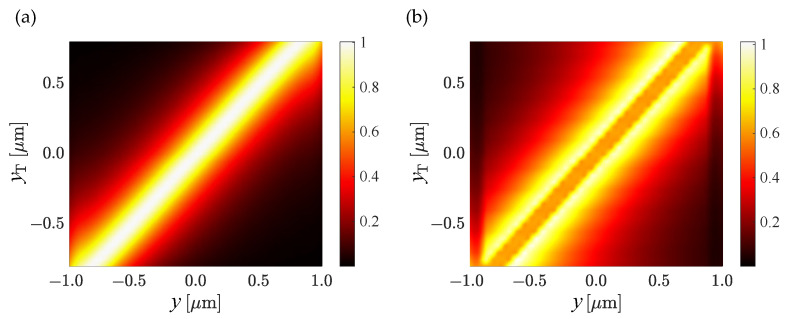
(**a**) TM_00_-mode profile ($\left|E_{x}\right|$) and (**b**) TE_00_-mode profile ($\left|E_{y}\right|$ ) along the *y*-direction as a function of the position $y_{T}$ of the trench.

**Figure 7 nanomaterials-13-00469-f007:**
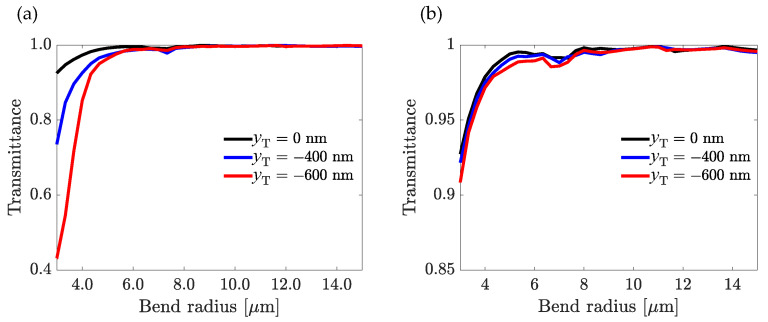
TM_00_ mode transmission after a U-turn bend for $y_{T}=0$ nm (black), $y_{T}=-400$ nm (blue), and $y_{T}=-600$ nm (red) for: (**a**) when $y_{T}$ is varied in the entire U-turn bend waveguide and (**b**) when $y_{T}=0$ is varied only at the bend section.

**Figure 8 nanomaterials-13-00469-f008:**
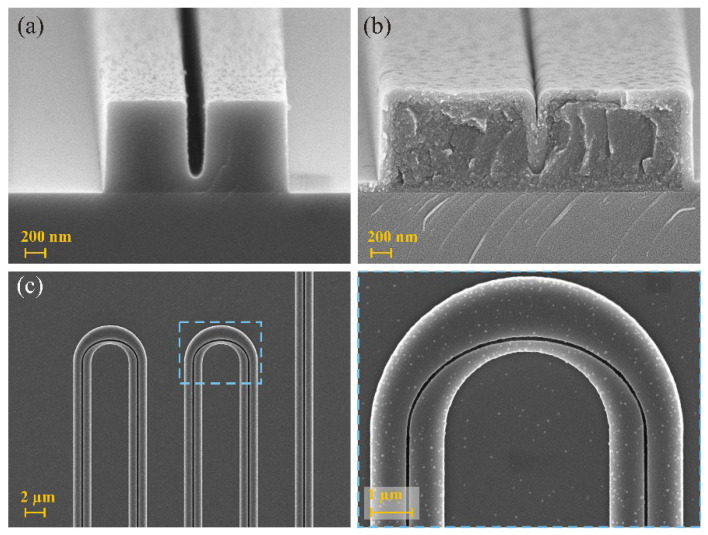
Scanning electron microscope pictures of examples of fabricated waveguide. (**a**) Cross section before ALD coating along the actual waveguide section. (**b**) Cross section after ALD coating along the tapered section of the waveguide. (**c**) Top view of a multi-bend waveguide with a zoomed-in section on the bend before TiO_2_ deposition.

**Figure 9 nanomaterials-13-00469-f009:**
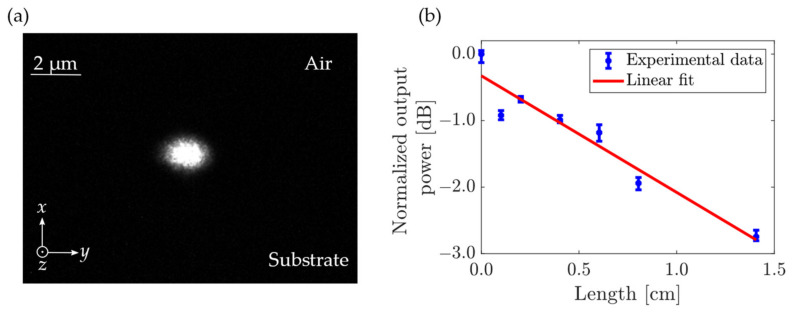
(**a**) Photograph of the output quasi-TM fundamental mode at the output of waveguide (chip 2) with bend radius of 5 µm. (**b**) Normalized power measured at the output of the waveguides (chip 1) as a function of the length of the waveguide (blue circles) and a linear fit (red line) using the cut-back method. Propagation losses are estimated at 1.75 ± 0.32 dB/cm.

## Data Availability

Not applicable.

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
