# Peer review of "Low Loss Vertical TiO2/Polymer Hybrid Nano-Waveguides"

_nanomaterials, 2023, doi:10.3390/nano13030469_

Round 1
Reviewer 1 Report
In "Low loss vertical TiO2/polymer hybrid nano-waveguides" the authors propose a new low loss polymer channel hybridized with a titanium dioxide core leading to a nano-waveguide. The authors have shown that a thin layer of material with a higher refractive index in a groove inside a waveguide core can increase confinement and reduce propagation loss. The authors call this device "nano-waveguides", although judging by the presented study, it is more suitable for the concept of "micro-waveguides", since less than 1000 nm, no studies are presented. Although it is necessary to call the cross section of the core is quite small. Nevertheless, the presented article is interesting and of scientific significance. In general, this manuscript causes a positive reaction and can be accepted for publication after minor revisions.
1. It seems to me that the authors were very carried away by the technical component in the introduction. It is necessary to indicate where such waveguides can be used, including whether they can be used in quantum technologies.
2. In section 3.1.3, the authors conducted the simulation, but it is not clear which models were used in this case. It is necessary to briefly describe how the simulation was carried out.
Author Response
We are thankful to Reviewer 1 for his/her positive comments and decision about our study. We agree on the fact that considering the entire waveguide cross-section, the waveguide is no longer a nano-waveguide. However, we decided to opt for the name “nano-waveguide” in comparison to silicon photonic nano-waveguides that are embedded in SiO2, for instance, and with a core of similar size than the one we present here. This choice comes from the fact that we offer a possible alternative to silicon photonics when wavelength or material compatibility impair.
- It seems to me that the authors were very carried away by the technical component in the introduction. It is necessary to indicate where such waveguides can be used, including whether they can be used in quantum technologies.
We agree on this point and thank the Reviewer for letting us know such a lack of opening. Technology is indeed the core driving force of our work, and we tend to forget at this stage of the development the possible applications. Although we made an opening in the conclusion of the paper, we improved here the introduction adding the following paragraph and the corresponding references. References 23, 24, and 25 have been added and the list of references has been updated.
Additional paragraph:
In this article we focus on the demonstration of the waveguide with a titania core. However, it is worth mentioning that as long as the refractive indices matches the condition , nothing prevent any change in the materials. One can imagine a multitude of applications for which hybrid waveguide is necessary but not easy at the wafer scale. For instance, one can include a gain medium within the trench in order to create an on-chip amplifier [23]. Another application could be the use of quantum dots or dyes inside the trench to provide up or down conversion of a signal in order to reach other wavelength ranges [24]. Emitters at very low concentration within the trench can provide very small amount of photon and with a proper design of the polymer section, a single photon source could be achieved [25] for quantum photonics purposes.
Additional references:
- Mu, J.; Dijkstra, M; Korterik, J.; Offerhaus, H.; García-Blanco, S.M. High-gain waveguide amplifiers in Si3N4 technology via double-layer monolithic integration. Photon. Res. 2020, 8, 1634-1641, doi.org/10.1364/PRJ.401055.
- Huang, X.; Cutinha, N.; Alcázar de Velasco, A.; Chandler, P.J.; Townsend, P.D. Upconversion in erbium doped YAG ion-implanted waveguides. Nucl. Instrum. Methods Phys. Res. B 1998, 142, 50-60, doi.org/10.1016/S0168-583X(98)00208-0.
- Papon, C.; Wang, Y.; Uppu, R.; Scholz, S.; Wieck, A.D.; Ludwig, A.; Lodahl, P.; Midolo, L. Independent operation of two waveguide-integrated single-photon sources, arXiv:2210.09826 [quant-ph] 2022, doi.org/10.48550/arXiv.2210.09826.
- In section 3.1.3, the authors conducted the simulation, but it is not clear which models were used in this case. It is necessary to briefly describe how the simulation was carried out.
We used the same mode solver than for the other (previous) simulations. We considered the cross section of the optimized waveguide and shifted the titania trench. The method is therefore similar, and no modifications has been made in the paper except mentioning it. However, we added the dimensions of the waveguide cross section.

Reviewer 2 Report
1. Page 4, line 138, "The width and height of each polymer section have been set to ? = 800 nm and ℎ =900 nm to ensure single modal behavior of the waveguide, when considering titania inside the trench." It need more detailed explanation that why choose 800nm/900nm.
2. Page 8, 3.2 fabricted sample. Author should firstly provide the designed parameters of waveguide, and then compare the designed and experimentally measured sizes of waveguide, especially the trench and the thickness of TiO2.
3. Figure 8, the scale bars of 8a and 8b are the same, 200 nm. As such, the sizes of waveguide in 8a and 8b seems to be different, which need authors to check.
Author Response
We thank Reviewer 2 for his/her positive and constructive comments.
- Page 4, line 138, "The width and height of each polymer section have been set to ? = 800 nm and ℎ =900 nm to ensure single modal behavior of the waveguide, when considering titania inside the trench." It need more detailed explanation that why choose 800nm/900nm.
We have indeed not given in the paper the detailed simulations for the polymer section, since it would have made the paper confusing. The choice for these dimensions is coming from a huge set of simulations in which w, h, and the trench size t have been varied systematically. It turned out that for a relatively broad range of couple (w, h), the waveguide remains single mode and t is the parameter with the most significance. Therefore, we focused on this one only in the manuscript. The choice for h and w comes from the fabrication constrains: the aspect ratio of the trench compared to the overall supporting polymer structure. We still have a lot of degree of freedom for these parameters, however, to limit the losses, we opted for the broader ones. A short paragraph has been added to the manuscript in order to clarify this point.
Added text:
It is to be noted that several solutions exist for the dimensions of the polymer sections to maintain a single mode waveguide. We opted for these dimensions for fabrication constraints and for a better mode confinement in the trench, for which the heigh plays the most important role.
- Page 8, 3.2 fabricted sample. Author should firstly provide the designed parameters of waveguide, and then compare the designed and experimentally measured sizes of waveguide, especially the trench and the thickness of TiO2.
Thank you for pointing out this omission. We added in the beginning of section 3.2 the target parameters and a short note about the difference to fabricated sample.
Added text:
The target parameters chosen for the fabrication of the structure are: w = 800 nm, h = 900 nm, and t = 100 nm. A less than 5% error on these dimensions was measured from the SEM pictures. Figure 8 presents several SEM pictures acquired on different samples.
- Figure 8, the scale bars of 8a and 8b are the same, 200 nm. As such, the sizes of waveguide in 8a and 8b seems to be different, which need authors to check.
We agree that at first sight this might be confusing. It turns out that the first SEM picture (8a) is taken before coating in the middle of the straight part of the waveguide. This section corresponds to the simulated optimized parameters. The second one (8b) has been taken within the taper, the cross-section of the waveguide is therefore larger than for 8a, but the magnification was identical. There is no mistake in the scale bars. We made the text clearer in the revised manuscript.
Added text:
Figure 8b is an SEM picture of the cross-section of the waveguide after TiO2 deposition (cleaved within the tapered section of the waveguide, which explains a larger waveguide than in Fig. 8a).
